# Distribution, Risk Assessment and Source of Heavy Metals in Mangrove Wetland Sediments of Dongzhai Harbor, South China

**DOI:** 10.3390/ijerph20021090

**Published:** 2023-01-07

**Authors:** Yuan Guo, Xianzhong Ke, Jingxian Zhang, Xinhui He, Qinghua Li, Yanpeng Zhang

**Affiliations:** 1School of Environmental Studies, China University of Geosciences, Wuhan 430078, China; 2Wuhan Center, China Geological Survey (Central South China Innovation Center for Geosciences), Wuhan 430205, China

**Keywords:** heavy metals, mangrove, distribution, contamination assessment, source

## Abstract

Heavy metals are common environmental contaminants that are toxic, non-biodegradable, and bioaccumulative. They can bioaccumulate through the food chain and present a risk to both public health and ecology. Therefore, this study takes the mangrove wetland of Dongzhai Harbor as an example. The concentrations of heavy metals such as As, Cd, Cr, Cu, Ni, Pb, and Zn in the surface sediments of mangrove wetlands were measured to reveal their distribution, the contamination level was assessed, and the sources of contamination were analyzed. The distribution of Cr, Zn, Ni, Pb, Cu, and Cd concentrations are: Yanfeng East River > Sanjiang River > Yanzhou River > Yanfeng West River, while the As concentration in the Yanfeng West River is greater than that in the Yanfeng East River. According to the correlation analysis, the concentrations of Cr, Zn, Ni, Cu, and Cd are significantly and positively correlated with total organic carbon (TOC), total phosphorus (TP), total nitrogen (TN), and salinity (SAL) and shared a significantly negative correlation with pH. There is moderate contamination risk of As and slight contamination risk of Cd, Cr, Cu, Ni, Pb, and Zn in most regions within the study area. Cd, Cr, Cu, Ni, Pb, and Zn exhibit the same sources, which are mainly influenced by human sources such as aquaculture, agricultural cultivation, and livestock farming, while the source of As comes from aquaculture.

## 1. Introduction

Mangroves are widely distributed in the upper intertidal zone of the tropics and subtropics, with high productivity and rapid sedimentation rates [1]. Under the influence of urbanization, industrialization, and other human activities, mangroves are facing the threat of heavy metal (HM) contamination [2]. Heavy metals (HMs) are common environmental contaminants that are toxic and non-biodegradable [3]. They can bioaccumulate through the food chain and present a risk to both human health and the ecosystem [4]. In the environment, HM contamination mainly refers to elements with significant biotoxicity like cadmium (Cr), mercury (Hg), lead (Pb), chromium (Cd), and metalloid arsenic (As). General HMs like copper (Cu), zinc (Zn), and nickel (Ni) are included as well [5].

The fine-grained sediments of mangrove wetlands are rich in organic matter and are prone to accumulating HMs from rivers and tidal waters, which is considered an important “HM sink” [6]. The special anoxic environment in the sediments allows HMs to re-enter the water through processes such as resuspension, redox reactions, and degradation of organic matter [7]. It makes the sediments an endogenous source of HM contamination, greatly increasing dissolved HM concentration in water. As a contamination purifier, mangroves can absorb, accumulate, and transfer HMs into the soil [8]. However, their withered leaves become part of the sediments and participate in HM accumulation as organic debris. Therefore, the migration and transformation of HMs between mangrove plants and sediments is a complicated and dynamic process [9,10].

HMs have attracted worldwide attention as one of the most serious potential human contaminations in mangrove ecosystems [3]. With the increasing focus on the protection and management of mangroves, the ecological effects of mangrove wetlands on HMs have received widespread attention. By analyzing the spatial distribution of HMs in mangrove sediments and their relationship with the physicochemical properties of sediments, the impact of mangroves on the absorption and migration of HMs can be effectively ascertained [11]. Moreover, heavy metal concentrations were significantly and positively correlated with organic matter content [11,12]. The vertical distribution of HMs in the sediments also varied significantly and was mainly enriched in the surface layer of the mangrove sediments. HMs will only migrate to lower sediment layers once the surface sediments are saturated [13].

Hainan Dongzhai Harbor Mangrove National Nature Reserve is the earliest established mangrove wetland with the most resources and the richest tree species in China [14]. In recent years, the mangrove wetland ecosystem of Hainan Island has been seriously damaged, and the ecological environment has deteriorated due to long-term unreasonable exploitation and activities of sea enclosing, deforestation, aquaculture, and reclamation [15]. The current research focuses on the spatial distribution and contamination assessment of HMs in the sediments of Dongzhai Harbor [16,17,18]. However, there is a lack of detailed studies in ascertaining the contamination sources, which is a great challenge to local environmental management and the protection of mangrove wetlands.

In this study, the distribution of HMs in the study area was analyzed based on the concentrations of As, Cd, Cr, Cu, Ni, Pb, and Zn in the mangrove wetland sediments of Dongzhai Harbor. Moreover, the factors affecting the distribution of HMs were also analyzed along with land use type. In addition, correlations between sediment geochemical indicators (pH, salinity (SAL), total organic carbon (TOC), total phosphorus (TP), and total nitrogen (TN)) and HM elements were analyzed. On this basis, the geo-accumulation index (*I_geo_*) and potential ecological risk index ($E_{r}^{i}$) were used to assess the level of contamination of HM. Principal component analysis was used to explain the sources of HMs. The purpose of this study is to comprehensively and systematically assess the level of contamination and the sources of HMs in the sediments of the mangrove wetland in Dongzhai Harbor and to offer a theoretical foundation for the management and protection of mangrove wetlands.

## 2. Materials and Methods

### 2.1. Study Area

Dongzhai Harbor (110°32′~110°37′ E, 19°54′~20°01′ N), a drowned valley harbor, located in the northeast of Hainan Island, has the largest continuous mangrove wetland in China [15]. The Zhuxi River, Yanfeng East River, Yanzhou River, and other small streams flow into it. Greatly affected by tides, it has an annual average tidal difference of 0.95 m and an average water area of about 55 km^2^ [19]. The landform of Dongzhai Harbor is mainly a coastal accumulation plain and consists of fine sand and powder-fine sand from the Holocene coastal phase mounding layer; having a tropical monsoon climate, Dongzhai harbor’s annual average temperature ranges from 23.3 to 23.8 °C [15,20]. The annual average temperature of the seawater surface is 25.5 °C [21]. The annual average sunshine duration is 2200 h, and it never snows [21]. The annual average precipitation is about 1676.4 mm, mainly in summer and autumn (Figure 1) [20].

### 2.2. Sampling and Analysis

Sampling sites were set up to collect surface sediment samples in the mangrove wetlands on both banks of the Yanfeng West River, Yanfeng East River, Sanjiang River, and Yanzhou River in Dongzhai Harbor, with 20 sampling sites from upstream to downstream in each river and a total number of 80 sampling sites. The sampling sites were set up by taking into account topography, hydrological patterns, mangrove density, settlement distribution, and tidal influence. Samples were collected during the summer low tide in August 2020. Each sample was collected from three points near the sample site and mixed, with a soil weight of at least 1 kg. Surface sediments (approximately 0–20 cm) were collected using a sediment Peterson grab sampler. Branches, leaves, and other debris were picked out and stored in polyethylene bags. The surrounding environmental conditions were recorded. The sampling apparatus was cleaned before each sampling.

The sediment samples were freeze-dried, ground, and passed through 0.25 mm and 0.075 mm sieves, respectively, and then stored at room temperature for backup. The test indexes were as follows: pH, SAL, TOC, TN, TP, As, Cd, Cr, Cu, Ni, Pb, and Zn. The pH, SAL, TOC, TN, and TP tests were done at the School of Environmental Studies, China University of Geosciences (Wuhan). The tests of As, Cd, Cr, Cu, Ni, Pb, and Zn were done at Aoshi Mineral Laboratory, Guangzhou. The dried sediments were extracted at a water-to-soil ratio of 1 to 5, and the pH and SAL of the sediments were measured in the supernatant using a portable water quality multi-reference meter (HACH, HQ30D, Loveland, CO, USA) [22]. TOC and TN were determined by a CHONS elemental analyzer (ELEMENTAR, Vario EL III, Langenselbold, Germany), and TP was determined by spectrophotometer (Hitachi, U-1800, Tokyo, Japan) after ashing at 550 °C in a muffle furnace (Jianing Instrument, SX2, Yantai, China) [22,23].

The HMs were tested as follows: firstly, 1 g of the sample was weighed into a Teflon tube and then the sample was pre-oxidized with 10 mL of aqua regia (volume ratio HNO_3_: HCl = 1:3, Analytical Reagent, Charlotte, NC, USA). Next, 10 mL of HF (Analytical Reagent) was added, and the reaction was heated in an oven (Jinghong Instrument, DHG-9246A, Shenzhen, China). The solution was then evaporated until nearly dry. After the residual solution was diluted with HF and fixed, Cd, Cr, Cu, Ni, Pb, and Zn were measured by inductively coupled plasma mass spectrometry (Thermo Fisher Scientific, iCAP RQ, Waltham, MA, USA), while As was determined by atomic fluorescence spectroscopy (Beijing Haiguang Instrument, AFS-9700, Beijing, China) [24]. After correction for spectral interferences between the elements, the final results were acquired. Standard sediment reference material (GBW 07314), provided by the State Oceanic Administration of China, was used to assess the precision and accuracy of the method. The relative deviation (RD) of the method is less than 10% for precision control and less than 10% for accuracy control (RE). Quality control and accuracy of results were ensured through repeat testing.

### 2.3. Assessment Methodology

#### 2.3.1. Geo-Accumulation Index

The geo-accumulation index (*I_geo_*), a method proposed by German scientist Muller in 1969 for studying the quantitative index of the contamination level of each HM in sediments, fully takes into account human contamination, environmental geochemical background concentrations, and factors of natural diagenesis [25]. The geo-accumulation index is calculated by the following formula:(1)$$I_{geo}=\log_{2}\left(\frac{C_{n}}{1.5B_{n}}\right)$$
where $C_{n}$ is the determined concentration of HM element n in the sample sediments and $B_{n}$ is the geochemical background concentrations of this HM element *n* in the sedimentary rocks [26]. Based on the value of *I_geo_*, the degree of HM contamination was divided into seven levels, as shown in Table 1.

#### 2.3.2. Potential Ecological Risk Index

The potential ecological risk index ($E_{r}^{i}$), a simple and intuitive method to assess HM contamination in sediments, is widely used in sediment contamination assessment [28,29,30]. The method takes into account the migration of HMs in sediments and also reflects the degree of contamination of HMs in the environment and the potential ecological risk to the aquatic environment. Its calculation formula is as follows:(2)$$E_{r}^{i}=T_{r}^{i}\times C_{f}^{i}=T_{r}^{i}\times C_{i}/C_{n}^{i}\;$$
(3)$$\mathrm{RI}=\sum E_{r}^{i}$$
where RI is the comprehensive potential ecological risk index of HM *i* in sediments; $T_{r}^{i}$ is the toxicity coefficient of HM element *i* in sediments, which is used to reflect the toxicity level of monomial HM and the sensitivity of organisms to pollutants [31]; $C_{i}$ is the measured concentration of HM element *i* in sediments; $C_{n}^{i}$ is the background concentration of HM element *i* in sediments [26]. Based on the value of $E_{r}^{i}$, the risk level was graded as in Table 2.

### 2.4. Data Processing and Analysis

SPSS 26.0 software was used for the HM concentration, correlation, and source. ArcGIS10.7 software was used to plot the spatial map of HMs. Origin2021 software was used to visualize the HM concentrations and contamination assessment results.

## 3. Results and Discussion

### 3.1. HM Concentrations in Sediments

The statistical results of HMs in the surface sediments of Dongzhai Harbor are shown in Table 3. The following rankings of the HM concentrations (range, mean concentrations, μg/g) were made in decreasing order: Cr (36–152, 86) > Zn (34–137, 82) > Ni (14.7–91.8, 41.3) > Pb (15.3–40.9, 26.5) > Cu (7.8–45.4, 21.3) > As (6.2–24.7, 13.7) > Cd (0.02–0.14, 0.07). Cr and Zn concentrations were relatively high compared to the other HMs, accounting for approximately 62% of the total concentrations of the 7 HMs in this study.

The coefficient of variation provides a simple indication of the distribution of HMs in the sediments. A higher coefficient of variation in concentration indicates a more heterogeneous distribution of HMs and a higher degree of dispersion. The following are the values of the coefficients of variation for the HMs: Cd (0.49) > Cu (0.44) > Ni (0.43) > Cr (0.35) > Zn (0.32) > As (0.21) > Pb (0.19). The value of the coefficient of variation of Cd, Cu, and Ni is larger than 0.36, indicating that their concentrations varied greatly in different areas. Comparing the HM concentrations of the sample with the background values of the Dongzhai Harbor sediments, the concentrations of Cd and Pb were 94% and 109% of the background values, respectively, suggesting that they were less influenced by humans [26]. However, the concentrations of As, Cr, Cu, Ni, and Zn were significantly higher than the background. The ratio of the average concentration of the elements to the background was as follows: As (568%) > Ni (308%) > Cr (233%) > Cu (166%) > Zn (147%). It indicates that Dongzhai Harbor may be contaminated with these HMs to some extent. Among them, the exceedance rate of As and Ni is high, indicating that the contamination of As and Ni may be relatively more serious. In summary, HMs were found to be enriched in different degrees in the sediments. In order to accurately ascertain the contamination degree of each HM, a subsequent comprehensive contamination assessment is required.

Table 4 shows the concentrations of HMs in the surface sediments of mangrove wetlands in some regions of the world. The concentrations of Cd, Cu, Pb, and Zn in the surface sediments of the study area are low, which are in a lower middle level among the global mangrove wetlands. The concentration of Cr is much higher than in other areas, which is presumably related to the high background of Cr in the study area. The concentration of Ni is also higher than that in the sediments of other regions in the table. There is little information about the concentration of As except that its concentration in the study area is higher than that in Guanabara Bay wetland in Brazil [32,33,34,35,36]. In general, compared to other areas, the HM concentration in Dongzhai Harbor is still at a moderately low level. It is probably related to the restriction in Hainan Province on the development of traditional industries that have a serious impact on the ecological environment.

### 3.2. Spatial Distribution of HMs

Figure 2 shows the distribution of HMs in the mangrove wetland in Dongzhai Harbor. The distribution of the HMs in this study area was extremely heterogeneous. Therefore, the distribution characteristics of the HMs in the surface sediments are discussed one by one. The average concentrations of As in the four rivers in this study area were ranked in descending order as follows: Yanfeng West River (17.46 μg/g) > Yanfeng East River (14.60 μg/g) > Sanjiang River (11.94 μg/g) > Yanzhou River (10.72 μg/g). The average concentrations of Cr were ranked in descending order as follows: Yanfeng East River (117.50 μg/g) > Sanjiang River (94.78 μg/g) > Yanzhou River (77.95 μg/g) > Yanfeng West River (58.65 μg/g). The average concentrations of Cu were ranked in descending order as follows: Yanfeng East River (32.77 μg/g) > Sanjiang River (21.62 μg/g) > Yanzhou River (17.33 μg/g) > Yanfeng West River (13.76 μg/g). The average concentrations of Pb were ranked in descending order as follows: Yanfeng East River (30.06 μg/g) > Sanjiang River (27.02 μg/g) > Yanzhou River (26.97 μg/g) > Yanfeng West River (21.74 μg/g). The average concentrations of Zn were ranked in descending order as follows: Yanfeng East River (30.06 μg/g) > Sanjiang River (27.02 μg/g) > Yanzhou River (26.97 μg/g) > Yanfeng West River (21.74 μg/g). The concentrations of Pb and Cd were close to the background, but the concentrations of the other five HMs exceeded the established standard.

According to the distribution of different land use types in Figure 1, it can show that there are a large number of aquaculture ponds, arable land, and construction areas around the Yanfeng East River, where chemical fertilizers and pesticides containing HMs can flow into the wetland directly with the tide, resulting in serious damage to the mangroves on both sides of the Yanfeng East River [37,38]. Farther away from the Dongzhai Harbor, the Yanfeng East River flows more slowly, creating worse exchange conditions between the seawater and the river. Aquaculture bait and excrement are directly discharged into the river, leading to an increase in organic matter concentration in the sediments and increasing the adsorption of HMs in the sediments. In contrast, the Yanfeng West River is close to the harbor and far from the residential and aquaculture land, where the hydrodynamic conditions are better and the contamination level is lighter. Therefore, it is presumed that the hydrodynamic conditions are one of the important factors affecting the distribution of HMs. The poor hydrodynamic conditions of the Yanfeng East River made it the most seriously polluted one by Cr, Cu, Ni, and Zn.

As shown in Figure 3, the high concentration areas of As were mainly located in the whole section of the Yanfeng West River, the upper reaches of the Yanfeng East River, the middle reaches of the Sanjiang River, and the estuary of the Yanzhou River. It is inferred that there are more aquaculture ponds near the Yanfeng East River and the estuary of the Yanfeng West River, and the direct discharge of aquaculture wastewater into the river will lead to more As contamination in these areas. The distribution of mangroves in the middle reaches of the Sanjiang River is sparser than the whole section, where the adsorption of As is weak. Additionally, there are many more aquaculture ponds on both banks, with the discharge of wastewater that leads to higher As concentrations. Arsenic contamination is also relatively serious in the estuary of the Yanzhou River due to its vulnerability to oyster aquaculture, sea routes, dockyard contamination, and aquaculture contamination.

The high-concentration areas of Cd are mainly located in the Yanfeng East River and the estuary of the Sanjiang River. Cd is usually derived from phosphate fertilizer and may be released into the sediments through the application of phosphate fertilizer [39]. There are a large number of aquaculture ponds and agricultural land around the Yanfeng East River and the estuary of the Sanjiang River. The use of phosphorus fertilizer in aquaculture production, such as shrimp ponds and oysters, and agricultural production may lead to high cadmium concentration in the above study areas. The high-concentration areas of Cr, Cu, Ni, Pb, and Zn are mainly in the Yanfeng East River, the upstream area of the Sanjiang River, and the upstream area and the estuary of the Yanzhou River. There are a large number of aquaculture ponds, agricultural land, and residential areas along the Yanfeng East River; the upper reaches of Sanjiang River and Yanzhou River are near the Sanjiang Farm, where there is a large amount of arable farmland. The estuary of the Yanzhou River is mainly affected by contamination from docks and the aquaculture industry. It is inferred that human activities such as aquaculture, agricultural cultivation, and livestock breeding lead to higher concentrations of Cr, Cu, Ni, Pb, and Zn in the above areas than in other regions.

### 3.3. Geochemical Indicators Affecting HMs

The concentration of HMs in sediments depends not only on natural transportation and human emissions, but also on a variety of factors, such as the surface patterns of the sediments, organic matter, mineral fraction, and depositional environment of the sediments [40]. The mangrove wetland intertidal zone is characterized by high temperature and humidity, high SAL and sulfur, and richness of organic matter. Sediment pH, SAL, and organic matter have important effects on HM mobility and transformation [41]. In order to ascertain the environmental factors affecting the concentration and distribution of HMs in the surface sediments of the study area, Pearson correlation analyses were conducted between the seven target HMs and the pH, SAL, TOC, TN, and TP in the sediments (Figure 4). Sediment organic matter is a HM ion complexing agent and has an adsorption effect on HMs. In general, sediments with higher organic matter concentrations accumulate more HMs accordingly [42]. All HMs, except As, were significantly positively correlated with TOC, presumably because organic carbon has a lot of phenolic hydroxyl and carboxyl groups and a high cation exchange capacity. HMs in the environment are easily removed from the water through surface adsorption, cation exchange, and chelation reactions with organic carbon to form metal–organic complexes, resulting in a significant correlation between HMs and TOC in sediments [12,42]. pH showed a significantly negative correlation for all HMs except As, indicating that pH has a significant effect on the distribution of HMs to the surface sediments and that the release of HMs increases with increasing acidity (decreasing pH) [43]. In addition, the redox potential of the environmental system is related to pH, in that a decrease in pH increases the redox potential and enhances the oxidation capacity. Therefore, the acidic environment benefits the leaching of HMs in the carbonate-bound organic matter and sulfide-bound states [44]. With the exception of Pb, the SAL and the concentrations of all HMs showed a strong and positive correlation, presumably due to the fact that the mixed zone of light and brackish water is conducive to the flocculation and deposition of suspended particles [45]. With the exception of As, TP and TN had a positive correlation with the concentrations of HMs, probably because phosphate and nitrate can reduce the migration transformation of HMs through ion exchange and precipitation of freshly generated, biologically active minerals with poor solubility [45].

### 3.4. HM Contamination Assessment

#### 3.4.1. Geo-Accumulation Index

The results of the geo-accumulation index (Figure 5) showed that in the Sanjiang River, there was no contamination to moderate contamination of Cr, Cu, and Ni, moderate contamination of As, and no contamination of Cd, Pb, and Zn; in the Yanfeng East River, there was no contamination to moderate contamination of Cr, Cu, and Zn, moderate contamination of As and Ni, and no contamination of Cd and Pb; in the Yanfeng West River, there was no contamination to moderate contamination of Ni, moderate contamination to strong contamination of As, and no contamination of Cd, Cr, Cu, Pb, and Zn; in the Yanzhou River, there was no contamination to moderate contamination of Cr and Ni, moderate contamination of As, and no contamination of Cd, Cu, Pb, and Zn. In summary, the contamination degrees of each HM in the surface sediments of Dongzhai Harbor mangrove wetland are ranked as follows: As > Ni > Cr > Cu > Zn > Pb > Cd. Cd and Pb are in the no contamination level, while their concentrations are 94% and 109% of the background values, respectively. The levels of Cd are lower than the background values, and the levels of Pb are higher than but very close to the background values, so it is reasonable for the *I_geo_* of both to be at the no contamination level. As, Ni, Cr, Cu, and Zn are all contaminated to varying degrees, with As showing the highest level of contamination, reaching moderate levels.

#### 3.4.2. Potential Ecological Risk Index

As shown in Figure 6, the $E_{r}^{i}$ of Cr in the sediments of the study area ranged from 1.86 to 7.86, with a mean value of 4.66; the $E_{r}^{i}$ of Cu ranged from 3.03–16.57, with a mean value of 8.30; the $E_{r}^{i}$ of Ni ranged from 5.19 to 32.42, with a mean value of 15.40; the $E_{r}^{i}$ of Pb ranged from 3.17 to 8.48, with a mean value of 5.44; the $E_{r}^{i}$ of Zn ranged from 0.59 to 2.39, with a mean value of 1.47; the $E_{r}^{i}$ of As ranged from 35.77 to 95.00, with a mean value of 56.77; the $E_{r}^{i}$ of Cd ranged from 9.09 to 63.64, with a mean value of 32.29. The $E_{r}^{i}$ of Cr, Cu, Ni, Pb, and Zn in the study area were all less than 40, which meant a low ecological risk. The $E_{r}^{i}$ of Cd in 75% of the sampling sites was less than 40, which meant a low ecological risk level. Only 25% of the sampling sites faced a moderate ecological risk level of Cd, most of which were located in the Yanfeng East River. In 80% of the sampling sites, the $E_{r}^{i}$ of As was between 40 and 80, which meant a moderate ecological risk level.

The results of the RI indicate that 40% of the sampling sites were at low risk, while 60% of the sampling sites were at moderate risk. At present, the mangrove wetlands in Dongzhai Harbor are at moderate and low risk. There is no sign of considerable and very high risk. Aquaculture and agriculture as anthropogenic sources of heavy metals can increase HM concentration in sediments [46,47]. The results of RI in the Yanfeng East River are significantly higher than other rivers, presumably due to its proximity to Yanfeng town, higher frequency of human activities, and denser distribution of aquaculture ponds and arable land. At the same time, mangrove plants have a sorption effect on heavy metals [46]. The mangrove density of the Yanfeng East River is lower than other rivers, resulting in lower functions of enriching and adsorbing HMs. Consequently, the HM contamination in the Yanfeng East River is more severe.

### 3.5. Sources of HMs

In order to visualize the proximity and similarity among the HMs, a variable clustering analysis was performed on the HMs in the surface sediments (Figure 7). The HMs in the surface sediments of the study area can be divided into two components. The first component consists of Cd, Cr, Cu, Ni, Pb, and Zn. The above HMs are more closely related and may have the same or similar sources. The second component is As, indicating that As may be different from other HMs and have other sources. The Kaisere–Meyere–Olkin (KMO) test and Bartlett’s spherical test (KMO of 0.723 > 0.6, *p* = 0 < 0.05) were performed with SPSS 26.0 to indicate that the raw data were suitable for factor analysis [42]. Based on the cumulative contribution of variance of HMs (Figure 8), the sources of HMs were explored by intercepting two principal factors with eigenvalues >1. The results were consistent with the conclusions obtained from cluster analysis. The cumulative contribution of the first two principal components amounted to 87.74% and was basically representative of the information contained in the data. The variance contribution rate of the first principal component (PC1) was 72.31%. Loads of Cd, Cr, Cu, Ni, Pb, and Zn were 0.781, 0.957, 0.975, 0.959, 0.848, and 0.971, respectively, indicating that the above six HMs had the same source. From the spatial distribution of HM concentrations, it can be seen that the high concentration areas of Cr, Cu, Ni, Pb, and Zn are mainly in the Yanfeng East River, the upper reaches of the Sanjiang River, the upstream area of the Yanzhou River, and the estuary of the Yanzhou River, while the high concentration areas of Cd were distributed in the Yanfeng East River. As the aquaculture, livestock, and poultry industries cause more environmental problems, some HMs remain in the Yanfeng East River. Previous studies have shown that aquaculture, such as shrimp ponds and oysters, is a source of Cd, Cr, and Cu [47,48]. The upper reaches of the Sanjiang River and Yanzhou River are close to Sanjiang Farm with a large amount of cultivated farmland. Agricultural activities are an important source of HMs. The irrational application of agricultural substances such as sewage irrigation and chemical fertilizers brings about long-term accumulations of Zn, Cd, and Pb. Pb contamination in soil caused by exhaust emissions from agricultural machinery using diesel or gasoline as the main fuel in the agricultural production process is non-negligible as well [49]. The estuary of the Yanzhou River is located in the East Tide Ditch of the Dongzhai Harbor Lagoon. It is mainly affected by oyster aquaculture, sea routes, and dockyard contamination, which may be one of the sources of Pb and Zn contamination [50]. In summary, PC1 is a mixed source of aquaculture, agricultural cultivation, and livestock farming.

The variance contribution rate of the second principal component (PC2) was 15.43%. Arsenic had a large positive load (0.975). The results of *I_geo_* and $E_{r}^{i}$ in this study showed that the ecological risk level of As was moderate in all river areas in the study area, and the high concentration areas were mainly distributed in the upper reaches of the Yanfeng West River and Yanfeng East River. The contamination of As in Dongzhai Harbor Mangrove Reserve mainly came from the sources of pesticides and fertilizers used extensively in agricultural production [51]. Therefore, PC2 is an aquaculture source.

## 4. Conclusions

(1) The distribution of HMs in the Dongzhai Harbor mangrove wetland is extremely uneven, with the concentrations ranked as follows: Cr > Zn > Ni > Pb > Cu > As > Cd. Among them, the high concentration areas of As are mainly distributed in the Yanfeng West River; the high concentrations areas of Cd are mainly distributed in the Yanfeng East River; the high concentrations areas of Cr, Cu, Pb, Ni, and Zn are mainly distributed in the Yanfeng East River, the upper reaches of the Sanjiang River, the upper reaches of the Yanzhou River, and the estuary of the Yanzhou River. In general, the concentration of HMs in the study area is still at a moderately low level compared with other regions in the world, which may be related to the restriction of Hainan Province on the development of traditional industries that have a serious impact on the ecological environment.

(2) The HM concentrations are closely related to the pH, SAL, TOC, TN, and TP of sediments. TOC is significantly positively correlated with Cr, Cd, Ni, Pb, Cu, and Zn. pH is significantly negatively correlated with Cr, Cd, Ni, Pb, Cu, and Zn. SAL is significantly positively correlated with As, Cr, Cd, Ni, Cu, and Zn. TP and TN are positively correlated with Cr, Cd, Ni, Pb, Cu, and Zn.

(3) The results of the *I_geo_* showed that there are As, Ni, Cr, Cu, and Zn contamination of different degrees, while there are no risks of contamination from Cd and Pb. The degree of As contamination is the highest, reaching a moderate level, and the potential ecological risk of As is moderate in 80% of the sampling sites. The potential ecological risk of Cr, Cu, Ni, Pb, and Zn contamination is low. The potential ecological risk of Cd contamination is low at 75% and moderate at 25% of the sampling sites.

(4) The sources of HMs in the sediments of the Dongzhai Harbor mangrove wetland can be divided into two categories. For Cr, Cd, Ni, Pb, Cu, and Zn, they are homogeneous and mainly originate from mixed sources such as aquaculture, agricultural cultivation, and livestock breeding. The source of As mainly comes from aquaculture wastewater, with bait and excrement contributing more to the enrichment of As.

## Figures and Tables

**Figure 1 ijerph-20-01090-f001:**
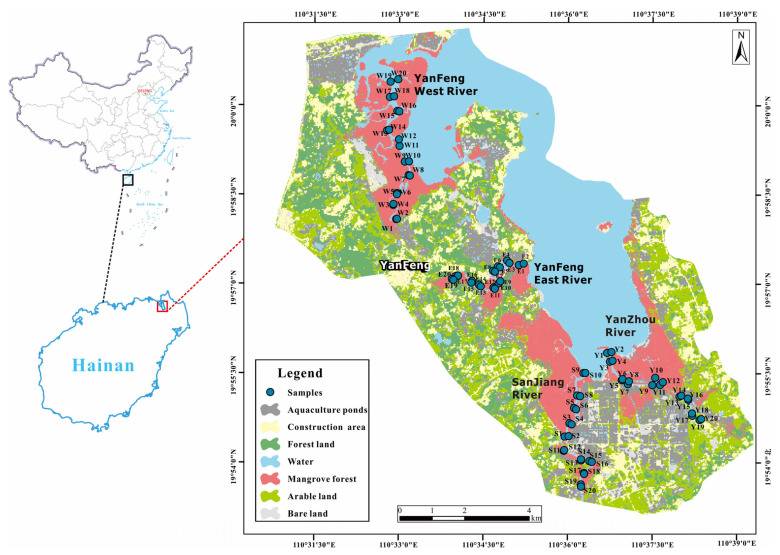
Sampling points of sediments in Dongzhai Harbor.

**Figure 2 ijerph-20-01090-f002:**
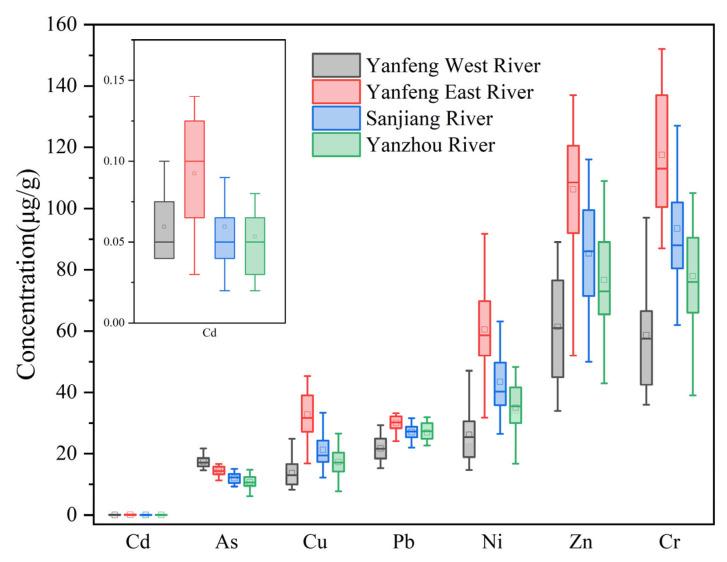
Distribution of HM concentration in sediments.

**Figure 3 ijerph-20-01090-f003:**
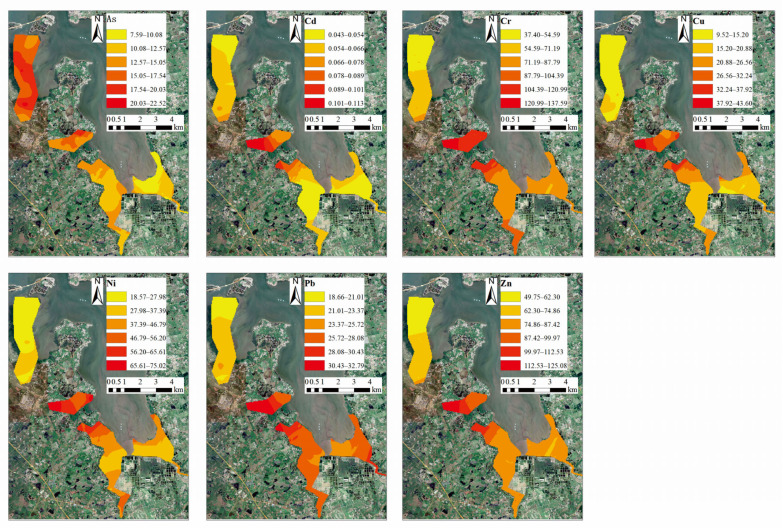
Spatial distribution characteristics of HMs in sediments.

**Figure 4 ijerph-20-01090-f004:**
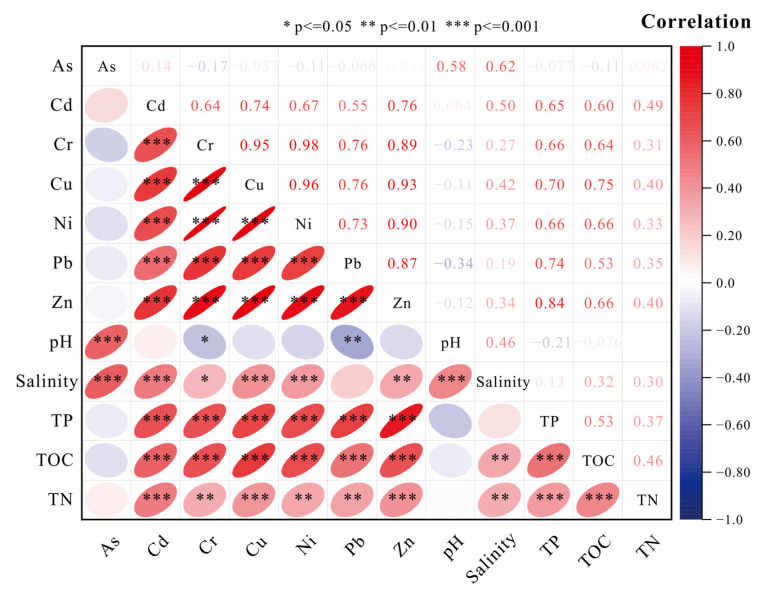
Correlation of HMs with geochemical indicators. Please refer to Table A1 (Appendix A) for the basic data of correlation.

**Figure 5 ijerph-20-01090-f005:**
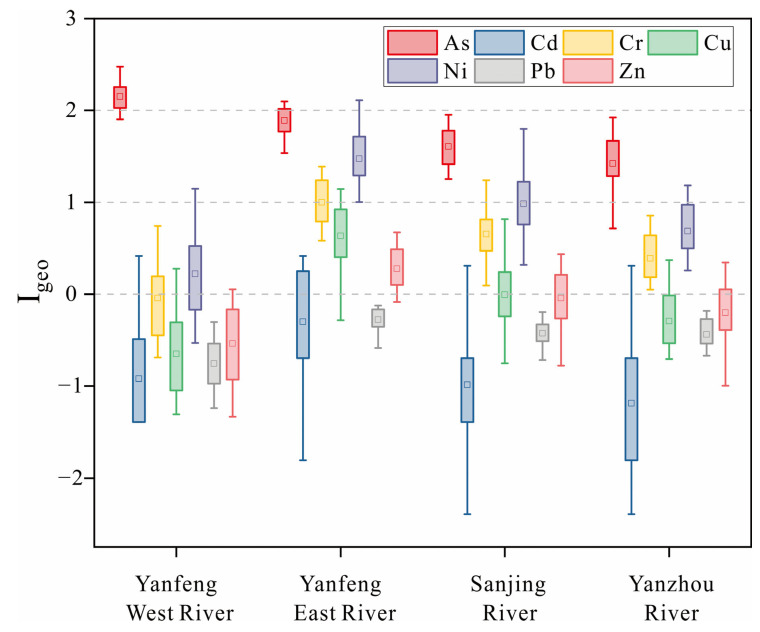
Risk assessment of geo-accumulation factor for HMs in sediments.

**Figure 6 ijerph-20-01090-f006:**
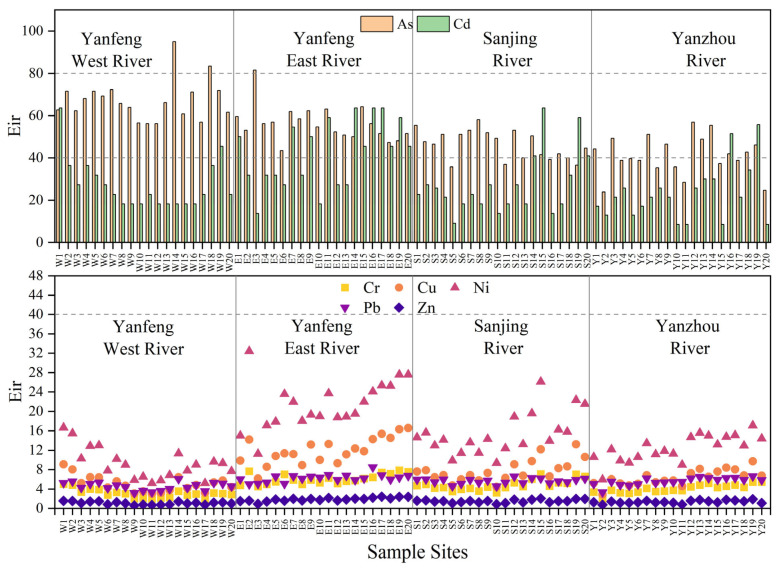
Potential ecological risk of HMs in sediments.

**Figure 7 ijerph-20-01090-f007:**
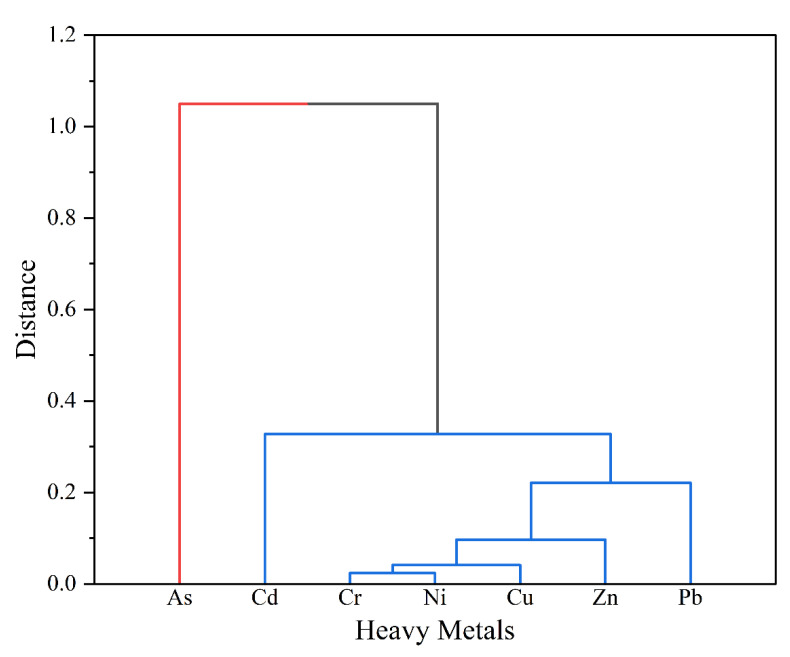
Cluster analysis of HMs in surface sediments in the study area.

**Figure 8 ijerph-20-01090-f008:**
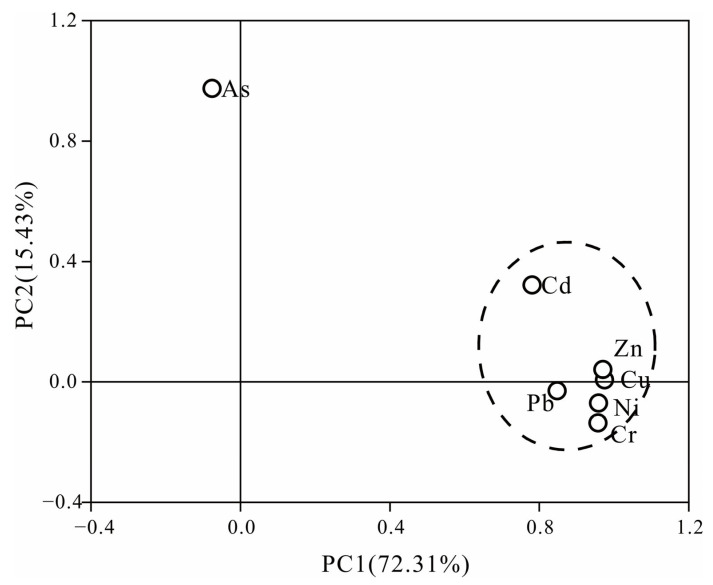
Principal component analysis of HMs in surface sediments in the study area.

**Table 1 ijerph-20-01090-t001:** Correspondence between geo-accumulation index and HM contamination level [27].

*I_geo_*	Level	Contamination Degree
*I_geo_* ≤ 0	0	No contamination
0 < *I_geo_* ≤ 1	1	No contamination to moderate contamination
1 < *I_geo_* ≤ 2	2	Moderate contamination
2 < *I_geo_* ≤ 3	3	Moderate contamination to strong contamination
3 < *I_geo_* ≤ 4	4	Strong contamination
4 < *I_geo_* ≤ 5	5	Strong contamination to extreme contamination
*I_geo_* > 5	6	Extreme contamination

**Table 2 ijerph-20-01090-t002:** Potential ecological risk assessment index for HMs and grading criteria [28].

$E_{r}^{i}$	Description	RI	Description
$E_{r}^{i}$< 40	Low ecological risk	RI < 150	Low risk
40 ≤ $E_{r}^{i}$ < 80	Moderate ecological risk	150 ≤ RI < 300	Moderate risk
80 ≤ $E_{r}^{i}$ < 160	Considerable ecological risk	300 ≤ RI < 600	Considerable risk
160 ≤ $E_{r}^{i}$ < 320	High ecological risk	RI ≥ 600	Very high risk
$E_{r}^{i}$ ≥ 320	Very high ecological risk	**–**	**–**

**Table 3 ijerph-20-01090-t003:** Descriptive statistics of HMs in this study area [26].

Sampling Area	As	Cd	Cr	Cu	Ni	Pb	Zn
Maximum (μg/g)	24.7	0.14	152	45.4	91.8	40.9	137
Minimum (μg/g)	6.2	0.02	36	7.8	14.7	15.3	34
Mean (μg/g)	13.7	0.07	86	21.3	41.3	26.5	82
Coefficient of variation/%	0.21	0.49	0.35	0.44	0.43	0.19	0.32
Background (μg/g)	2.6	0.07	39	13.7	14.2	24.1	57
Over background (%)	568	94	233	166	308	109	147

**Table 4 ijerph-20-01090-t004:** HMs in the sediments in other regions with mangrove (μg/g).

Country (Region)	As	Cd	Cr	Cu	Ni	Pb	Zn	References
Mai Po, Hong Kong	-	2.62	39.2	78.5	25	79.2	240	[32]
Quanzhou Bay, Fujian	-	0.181	68.1	42.5	28.1	12.28	184	[33]
Sungei Buloh, Singapore	-	0.181	16.61	7.06	-	12.28	51.24	[34]
Port Jackson, Australia	-	-	-	62–102	-	180–443	145–351	[35]
Guanabara Bay, Brazil	1.28	1.32	42.4	98.6	-	160.8	483	[36]
Mazatlan, Mexico	-	3.2	-	36	-	51.5	263.5	[36]

## Data Availability

The data that support the findings of this study are available from the corresponding author, upon reasonable request.

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
