# Peer review of "Distribution, Risk Assessment and Source of Heavy Metals in Mangrove Wetland Sediments of Dongzhai Harbor, South China"

_ijerph, 2023, doi:10.3390/ijerph20021090_

Round 1
Reviewer 1 Report
Dear Authors, in attached file you can found comments.

Reviewer 2 Report
The paper "Distribution, risk assessment and source of heavy metals in mangrove wetland sediments of Dongzhai Harbor, South China" submitted to me for evaluation is of a typical monitoring nature. Covers the issue of metal content: As, Cd, Cr, Cu, Ni, Pb and Zn in the surface sediments of mangrove wetlands of Dongzhai Harbor. The works were carried out in 2020 in the summer.
The tests performed are valuable from the point of view of assessing the quality of the aquatic environment. Bottom sediments are one of the components of the environment, which undoubtedly affects the quality of life of organisms associated with the bottom of reservoirs. However, everything depends on the incoming pollutants.
The results of these one-off studies are presented in several charts and tables. This is a database for this area and in the future it may be a reference for further research in this area.
In general, the work is valuable and, after making corrections, it can be the subject of further processing.
My comments below:
The authors write about the tests performed: pH, salinity, TOC, TN, TP and it is described in the methodology and correlations are given. However, the basic data in the tables is missing here. It would be worth posting them.
In the text of the work, references to literature, e.g. [1], should be placed not in superscript but normally in the text.
There is no reference to Figure 1 in the text. Please complete it.
[84 - 91] - reference to publication number [20] does not bring much, because this publication also does not specify the source where these values ​​come from.
[92] - Was reference material for sediments used? If so, what kind? Name, producer.
[101] - please provide the name of the sludge scooper or write that it was of your own production.
[113] and [114] - please enter the name of the country - manufacturer of the research equipment.
[114] – please enter the name of the equipment (spectrophotometer) and the name of the country – the manufacturer of the equipment.
[117 – 118] – what were the proportions, amounts and concentrations of the acids used.
[120] – "in oven" - please enter the name of the equipment.
[132] - it is worth specifying where the geochemical background for metals comes from (literature - geochemical maps, tables, who is the author). [133 - 134] - if Müller 1969 is quoted, why is this publication not included in the bibliography, but there is another one [21]. [137] – is formula 1 really correct? Missing 1.5 with Bn? [142] - enter a reference to the literature. [155] - enter a reference to the literature. [173 - 174] - no reference to table 3. [177] - reference to literature [21] should be before the period at the end of the sentence. [181 - 182] – values ​​are given with " . " and in table 3 without " . ", why? [188 – 198] - no reference to the literature. Figure 2 should be placed below text [204], not above it. Figure 3 should be placed below text [238], not above it. Bibliography: What does the [J] after article titles mean? I don't see this type of designation in the editorial requirements. Why are the authors' names capitalized? The publication year should be written in bold. The name of the journal and the number should be written in italics. Page numbers - not " : " before them, but " , ".
[497] – page numbers are not given.
Round 2
Reviewer 1 Report
Dear Authors, I accept your manuscript in this form. Kind regards, Reviewer